# Distinctions between Choroidal Neovascularization and Age Macular Degeneration in Ocular Disease Predictions via Multi-Size Kernels ξcho-Weighted Median Patterns

**DOI:** 10.3390/diagnostics13040729

**Published:** 2023-02-14

**Authors:** Alex Liew, Sos Agaian, Samir Benbelkacem

**Affiliations:** 1Department of Computer Science, Graduate Center of City University New York, 365 5th Ave., New York, NY 10016, USA; 2Robotics and Industrial Automation Division, Centre de Développement des Technologies Avancées (CDTA), Algiers 16081, Algeria

**Keywords:** optical coherence tomography, binary patterns, weighted median filter, macular disease, age-macular degeneration

## Abstract

Age-related macular degeneration is a visual disorder caused by abnormalities in a part of the eye’s retina and is a leading source of blindness. The correct detection, precise location, classification, and diagnosis of choroidal neovascularization (CNV) may be challenging if the lesion is small or if Optical Coherence Tomography (OCT) images are degraded by projection and motion. This paper aims to develop an automated quantification and classification system for CNV in neovascular age-related macular degeneration using OCT angiography images. OCT angiography is a non-invasive imaging tool that visualizes retinal and choroidal physiological and pathological vascularization. The presented system is based on new retinal layers in the OCT image-specific macular diseases feature extractor, including Multi-Size Kernels ξcho-Weighted Median Patterns (MSKξMP). Computer simulations show that the proposed method: (i) outperforms current state-of-the-art methods, including deep learning techniques; and (ii) achieves an overall accuracy of 99% using ten-fold cross-validation on the Duke University dataset and over 96% on the noisy Noor Eye Hospital dataset. In addition, MSKξMP performs well in binary eye disease classifications and is more accurate than recent works in image texture descriptors.

## 1. Introduction

Age-related macular degeneration is a visual disorder caused by abnormalities in a part of the eye’s retina and is a leading source of visual impairment Ref. [1]. Therefore, early diagnosis and treatment is critical Ref. [2]. Recently, retinal optical coherence tomography (OCT) images have been used to attain information regarding the health of the posterior eye (e.g., the retina and choroid) Ref. [3]. OCT is a quick, non-invasive medical imaging tool that uses low coherence interferometry to produce cross-sectional images of the retina and optic nerve head (ONH), or the most anterior part of the visual pathway, from the retina to lamina the cribrosa, which assess visual disorders, such as optic nerve disease, qualitative and quantitative. High-resolution (in µm range) scanner techniques such as optical coherence tomography (OCT) produce three-dimensional cross-sectional images of the eye’s biological tissues to visualize the individual layers of the posterior segment of the eye, allowing the diagnosis and monitoring of ocular diseases and anomalies Ref. [4]. This development of image acquisition reduces the cost of storage, which allows ophthalmologists to utilize these images to diagnose various eye diseases Refs. [5,6]. However, ophthalmologists would manually interpret each OCT image in the volumes to make a diagnosis decision. The increased data makes manual interpretation of the OCT volumes time-consuming Ref. [7].

Recent research in academia and industry has allowed artificial intelligence, Machine Learning, and Deep Learning (DL) to develop computerized algorithms to classify retinal disorders. These retinal disorders include diabetic macular edema (DME), aged macular degeneration (AMD), and Choroidal Neovascularization (CNV), see Figure 1. DME is the cause of glaucoma, where fluids or glucose build-up within the retinal layer causes vision impairment. Age-related diseases such as Age Macular Degeneration are caused by a build-up of drusen particles within the retinal epithelium layer. Dry AMD, which is usually diagnosed in older individuals, may cause daily life disruptions because it impairs the patient’s central vision. CNV, wet AMD, is when unusual blood vessels grow into the retina layers, causing fluids to leak and making the retina wet. For some people who are diagnosed with AMD, too many vascular blood vessels are produced. These new blood vessels spear from the choroid and then extend into the retina layers. These vessels are also leaky, allowing fluids and blood with red blood cells to enter the retina layers, distorting their vision. For this reason, early diagnosis and automated detection are essential in treating AMD Refs. [6,8,9].

The correct detection, precise location, classification, and diagnosis of choroidal neovascularization (CNV) may also be challenging if the lesion is small or if OCT images are degraded by speckle noises. Speckle noises in OCT images must be addressed to achieve good classification performance. Speckle noise, called multiplicative noise, is a granular noise texture. It degrades image quality as a consequence of interference among wavefronts in coherent imaging systems, such as radar, laser imaging, medical ultrasound, and optical coherence tomography. The speckle noise is signal-dependent and governed by the Fisher-Tippett distribution. Mathematically, Speckle noise Refs. [10,11,12,13] is expressed as *u*(*x*,*y*) = *v*(*x*,*y*) + *v*(*x*,*y*)*η*(*x*,*y*), where (*x*,*y*) are pixel position, *v*(*x*,*y*) is the clean image, *u*(*x*,*y*) is the noisy image, *η* is a Gaussian noise distribution with zero-mean and some variance *σ*^2^. It is well known that smoothing filters can reduce speckle noise. These techniques convolve an image with various neighbor sizes, 3 × 3, 5 × 5, etc., using a weighted average or selecting a median value, i.e., a non-weighted median filter. However, selecting a median value without any weights from a high noise-density image is not practical because a noise pixel may be selected instead of an image pixel. Therefore, we propose a new texture descriptor, MSKξMP, which echoes or repeats pixel values in a kernel to encode OCT images while avoiding a high level of speckle noises. This paper makes the following contributions:-Develops a new, simple, and highly accurate local texture descriptor algorithm, Multi-Size Kernels ξcho-Weighted Median Patterns (MSKξMP), to
(i)Avoid speckles noises;(ii)Perform eye disease classifications the Choroidal Neovascularization and Aged Macular Degeneration;(iii)Perform highly accurate eye disease classifications between Diabetic Macular Edema, AMD, and Normal eyes.
-Offers a Unique Singular Value Decomposition and Neighborhood Component Analysis based weighted feature selection method for establishing the optimal accuracy using SVM and Random Forest classification techniques.-Presents computer simulation results that show: (i) 99.78% accuracy on the Duke Dataset, 96.63% and 88.51% on the noisy Noor OCT Volume datasets; (ii) good eye disease diagnosis and recognition outcomes compared to the recent texture descriptors.

This paper has the following structure: the next section contains a Literature Review that offers an overview of methods that give detailed descriptions of the MSKξMP algorithm; the computer simulations section gives numerous experiments showing the performance of MSKξMP; finally, the conclusion and future work are presented in the last section.

## 2. Literature Review

In recent years, deep learning networks can perform vision recognition in various applications such as self-driving cars, natural language and image processing, and medical diagnosis (e.g., ocular diseases). Ref. [7] proposes an algorithm imitating how ophthalmologists form diagnoses by focusing on the information provided by OCT images during the classification process. This process is called a B-scan attentive convolutional neural network (BACNN). A self-attention module is used to cumulate features based on their clinical importance to obtain feature vectors. Ref. [1] proposes a multipath CNN architecture for the diagnosis of AMD. This architecture has five convolutional layers to classify AMD or normal images.

The multipath convolution layers extract critical global structures with a large filter kernel and use a sigmoid function as the classifier. Ref. [14] presented a CNN based on surrogate-assisted classification that classifies retinal OCT images automatically. Initial preprocessing was performed on each image using image denoising, thresholding, and morphological dilation to locate binary masks of the retina regions. The preprocessed images were employed to produce surrogate images in image augmentation, which were used to train the CNN model.

Several hand-crafted feature techniques show good classification performance on ocular disorder classifications in OCT images. Ref. [4]’s algorithm extracts features using multiscale histograms of oriented gradient descriptors and are classified using a support vector machine. Ref. [15] proposes the automated detection of AMD and DME from retina OCT images based on sparse coding, spatial pyramids, global representations, and dictionary learning. This process is coupled with preprocessing and a support vector machine classifier. Ref. [6] proposed a multiclass model for detecting AMD, DME, and normal using linear configuration patterns (LCPs) to extract pyramid and multiscale features. Recently, [10] proposed a texture descriptor called Alpha Mean Trimmed local binary patterns (AMT-LBP) based on Alpha Mean Trimmed Filter. The AMT-LBP encodes image pixels while partially avoiding and exploiting speckle noises. Table 1 summarizes the rest of the other recent techniques in ocular disorder classification.

The current works using hand-crafted features fall short compared to the deep learning technique in achieving extremely high classification accuracies. However, deep-learning models usually require long training times and heavy computational hardware such as GPUs. They are also data-hungry and complex. A simple, aggressive machine-learning approach is proposed here to overcome these shortcomings. This technique contains a smaller number of weights, making them easier to implement, requires less training time, and does not require specialized hardware Ref. [19]. Textural information is vital in analyzing eye diseases. Thus, this paper presents a novel texture descriptor, Multi-Size Kernel ξcho-Weighted Median Patterns, to distinguish between the various ocular diseases while achieving very high classification accuracies.

## 3. Methods

Our algorithm has the following steps. 

(i)Preprocessing: performs image segmentation using thresholding, then flattens and aligns the retina layers;(ii)Generating Hand-Crafted Features: generating hand-crafted features using Multi-Size Kernels Echo-Weighted Median Patterns (MSKξMP);(iii)Feature Weights Selection and Classification: best features are selected using Singular Value Decomposition, and neighborhood component analysis (NCA) is classified using Gaussian and Polynomial Kernels Support Machine Vector, Random Forest, Naïve Bayes, Adaboost, and RUS Boost.

### 3.1. Preprocessing

OCT scanner misalignment between the eye and sensor during the acquisition of the retinal images cause white areas in the image. Therefore, preprocessing Ref. [10] starts with removing white areas by assigning white pixels to black pixels. Then, the image is resized from a square image to a rectangular 256 × 512 image. Next, the image is binarized using Otsu thresholding Ref. [20], which generates a binary image (black and white), and a non-weighted median filter is applied afterward. The purpose of this median filter is not to remove noise in the image but to remove white areas in the binary image outside the retinal regions. A morphological dilated operator with a large structuring element is used on the median filtered image to close fluid region structures in DME and CNV images. Finally, the retinal layers are flattened by applying the polynomial fit of either the 2nd order or 3rd order based on the R^2^ fitness function. In most cases, the 3rd-order polynomial is selected to flatten the retinal layers.

### 3.2. Hand-Crafted Features

This section presents new, so-called Multi-Size Kernels ξcho-Weighted Median Patterns (MSKξMP), hand-crafted features. This is an extension of the Local Binary Patterns (LBP) feature extractor [21] and is used in many applications such as texture analysis, face recognition, object detection, fault diagnosis, and image retrieval. The advantages of LBP are that it is computationally simple, efficient, and is invariant to illumination. However, the disadvantages of LBP are its production of artifacts and noises, which distort the central pixel value causing classification degradation. To improve the robustness of LBP, different modifications of LBP were proposed.

In this proposal, we start with the generalized form of the LBP given by the following:(1)$$LBP\left(I_{C}\right)=\sum_{i\epsilon R}f\left(i\right)s\left(I_{i}-I_{C}\right)$$
where *I_C_* is the center pixel, *I_i_* are the surrounding pixels, *f*(*i*) is equal to 2*^i^* and *R* is a region defined by the kernel size. The kernel size is usually selected to be 3 × 3; however, sizes such as 5 × 5 and 7 × 7 are also known to be used. *s*(*I_i_* − *I_C_*) = {1, *I_i_* − *I_C_* ≥ *T*; 0, *I_i_* − *I_C_* < *T*} where *T* is selected to be zero. The generalized LBP operates on all the pixels in an image using a specific kernel size. Each of these kernels is placed over a pixel, *I_C_*, and is compared to its surrounding neighboring pixels *I_i_* using Equation (1). If a neighbor pixel is greater than or equal to the center pixel, then *s*(·) is assigned 1; if a neighbor pixel is less than the center pixel, then *s*(·) is assigned 0. A binary sequence is obtained, and each sequence is assigned to the appropriate decimal weight, 2*^i^*, which is then converted into a decimal value. The decimal weights, 2*^i^*, are then summed and encoded to structural information, generating an LBP image. The LBP is simple, fast, easy, and robust to illumination changes. However, it suffers when speckle noises are present.

The median LBP Ref. [22] was proposed to address these noises. The median LBP can be implemented by replacing *I_C_* in (1) with the median value of all the pixels within the region R: *I_C_* = *I_median_* = median(*R*) = median(*I*_0_, *I*_1_, *I*_2_, … *I_N_*_−1_), where *N* is the number of pixels in region *R*. However, the current median LBP does not go far enough to prevent speckle noises from falling into the texture-based calculation. For example, Figure 2a illustrates a 3 × 3 kernel with the following values [135, 75, 75; 135, 75, 75; 160, 135, 135]. The 75’s are image pixels without noise and the 135’s and 160’s are noisy pixels. When we select the median value within the 3 × 3 kernel, we obtain 135, a noisy pixel. To overcome this deficiency, Multi-Size Kernels ξcho-Weighted Median Patterns are proposed.

### 3.3. Multi-Size Kernels Echo-Weighted Median Patterns

MSKξMP is a textural feature descriptor similar to median LBP. MSKξMP encodes the textural pattern information by comparing surrounding pixels to the median pixel *I_median_*. However, this median value is calculated differently than Ref. [22]. Its surrounding pixels are defined using six different kernel sizes represented using *R_n_*_×*m*_, where *n*’s and *m*’s have the following values: 3 × 3, 5 × 5, 7 × 7, 3 × 5, 5 × 7, and 3 × 7. Numerous kernel sizes are used because each kernel captures a variation of texture within the OCT; one kernel can miss a specific feature that another kernel size picks up. A larger kernel like the 7 × 7 can capture regional details that a 3 × 3 kernel can miss and vice versa.

The MSKξMP is calculated in three steps for each kernel size, *R_n_*_×*m*_. The first is to calculate *I_median_*, the second is to find each *m_i_*’s found in *R_n_*_×*m*_, and the third step is to calculate each *medianLBP_n_*_×*m*_ using parameters obtained from steps one and two. The *medianLBP_n_*_×*m*_ is defined by the following equation:(2)$$medianLBP_{n\times m}\left(I_{C}\right)=\sum_{i\epsilon R_{n\times m}}f\left(i\right)s\left(m_{i}-I_{median}\right)$$
where *I_C_* is the current pixel being encoded and is the center pixel of the kernel. The combination of all *medianLBP_n_*_×*m*_ are the MSKξMP. The calculation of *I_median_* is motivated by the weighted median filter in Ref. [23], which *echoes* or repeats pixel values a predetermined number of times. Figure 2b illustrates the 3 × 3 kernel, but now echo-weights are applied. The integer of each weight indicates the number of repetitions, e.g., the second element, 75, is repeated two times, or the middle element, 75, is repeated three times. After applying the echo weight to the 3 × 3 kernel, the median was determined to be 75 or *I_median_* = 75, which is not a noise pixel. These weights can be determined by centering a two-dimensional Gaussian function onto *I_C_*, defined by the following:$$G\left(x,y\right)=\frac{1}{2\pi \sigma_{x}\sigma_{y}}e^{\frac{-\left(x^{2}+y^{2}\right)}{2\sigma_{x}\sigma_{y}}}$$
where *σ_x_* and *σ_y_* are standard deviations in the x and y directions, respectively. When a square kernel is utilized, the standard deviations should be the same, *σ_x_* = *σ_y_*, and when a rectangular kernel is utilized, the standard deviations could be different, *σ_x_* ≠ *σ_y_*. The selection of *σ_x_* and *σ_y_* is flexible, as long as the center pixel has the most weight and the surrounding weights are not too small compared to the center. Please see Figure 3, “MSKξMP Weights”. Notice the center weight is only one value higher than its neighboring values. The Gaussian Function is then normalized by its minimum value, *min*(*G*(*x*,*y*)), to ensure its minimum value equals 1. We then took the ceiling of each element in the normalized Gaussian Function to obtain integer values (or repetition values). The Echo formula can be represented by the following:
(3)$$\xi \left(x,y\right)=\lceil \frac{G\left(x,y\right)}{min\left(G\left(x,y\right)\right)}\rceil ,$$
where ⌈·⌉ indicates the ceiling operator, and *ξ*(*x*,*y*) represents the echo weights of our kernels. The standard deviations utilized in the 3 × 3 weight kernel, shown in Figure 3, are *σ_x_* = *σ_y_* = 1.2. The values of the standard deviations should increase with increasing kernel size. The weight of the center pixel should always have the highest repetition because this gives a non-noise pixel a better chance of being selected as the median.

Figure 3 shows MSKξMP images associated with their echo weights for all the kernel sizes used in this paper. Notice that the MSKξMP images with weights 5 × 7 and 7 × 7 had the coarsest texture, meaning there were more texture variations and they would provide the highest performance. For example, the MSKξMP of the CNV images can extract drusen particles better than the Fibonacci patterns. The DME image had the most noise, however, the 3 × 3 weighted MSKξMP image could encode less noise compared to the classical LBP.

Finally, *I_median_* can be written in terms the pixel intensities in *R_n_*_×*m*_:(4)$$I_{median}=median_{n\times m}\left(\varepsilon_{0}△I_{0},\varepsilon_{1}△I_{1},\;\ldots ,\varepsilon_{N-1}△I_{N-1}\right)$$
where △ is a duplication operator, *ε*_0_, *ε*_1_, … *ε_N_*_−1_ are elements in the matrix *ξ*(*x*,*y*) and *I_i_*, *i* = 0, 1, …, *N* − 1 are the pixel values in *R_n_*_×*m*_. $\varepsilon_{0}△I_{0}$ reads the following: *I*_0_ is duplicated *ε*_0_ number of times.

The second step is to determine *m_i_* values. These median values are selected within sub-regions of *R_n_*_×*m*_ and are color-coded in red, yellow, and blue, see top of Figure 4. The pair of regions to the left and right of the center pixel are coded in yellow, the corner sub-regions are coded in red, and the top and bottom sub-regions are coded in blue. A sample calculation of *m_i_*’s within a 5 × 5 kernel is illustrated in the following. East (yellow): *m*_1_ = median(0, 135) = 67.5, northeast (red): *m*_2_ = median(123, 147, 0, 135) = 106; north (blue): *m*_3_ = median(106, 123, 123, 106, 0, 0) = 106; northwest (Red): *m*_4_ = median(106, 106, 106, 106) = 106; West (yellow): *m*_5_ = median(106, 106) = 106; southwest (red): *m*_6_ = median(70, 70, 70, 70) = 70; south (blue): *m*_7_ = median(70, 153, 153, 70, 153, 153) = 153; and southeast (red): *m*_8_ = median(153, 128, 153, 128) = 140.5.

Once all the *m_i_*’s are calculated, we moved to the third step, where we compared *m_i_*’s to *I_median_* found in the first step using (2). The comparison encoded the *m_i_*’s to binaries which were then multiplied by their respective decimal weights, and the sum was calculated to obtain *medianLBP5 × 5*(*I_C_*). The rest of the *medianLBP_n_*_×*m*_’s are obtained similarly. Figure 3 shows that each *medianLBP_n_*_×*m*_ image displays different textural information, which is beneficial in distinguishing the different types of ocular disorders.

Four histograms were generated for each *medianLBP_n_*_×*m*_ image to extract their features. One histogram was generated from the entire *medianLBP_n_*_×*m*_ image, which represented the global features. The other three histograms were generated from local overlapping regions and labeled with different color brackets, as seen in Figure 4. Each local regions had a size of 256 × 256 to capture local features. The four histograms from each *medianLBP_n_*_×*m*_ are concatenated to form the MSKξMP feature vector for each OCT image.

## 4. Computer Simulation

### 4.1. Datasets

Three datasets were used to test the performance of MSKξMP. The first dataset, designated as Dataset 1, was taken from Duke University, Harvard University, and the University of Michigan Ref. [4]. This dataset consisted of SD-OCT volumetric scans acquired from 45 patients: 15 normal patients, 15 patients with dry AMD, and 15 patients with DME. All SD-OCT volumes were acquired in Institutional Review Board-approved protocols using Spectralis SD-OCT (Heidelberg Engineering Inc., Heidelberg, Germany). The second dataset, which will be designated as Dataset 2, was acquired from the Noor Eye Hospital dataset Ref. [9] and consisted of 148 SD-OCT volumes (48 AMD, 50 DME, and 50 Normal), acquired by using the Heidelberg SD-OCT imaging system at Noor Eye Hospital in Tehran (NEH). Each volume consisted of 19 to 61 B-scans; the resolution of the B-scans was 3.5 μm, and the scan dimension was 8.9 × 7.4 mm^2^.

The third dataset, Dataset 3, was also collected by the Heidelberg SD-OCT imaging system at Noor Eye Hospital (NEH) and was obtained from the Mendeley database website Ref. [8]. It contained 16,822 OCTs images of Normal (120 volumes), Drusen (160 volumes), and CNV (161 volumes). However, 12,641 images (3234 CNV, 3740 Drusen, and 5667 Normals) were selected for our experiments. This is because we were only keeping worst-case condition images for each volume, whereas if a patient was labeled a CNV case, only CNV B-scans within that volume were included for training and testing procedures. Normal and drusen B-scans of that patient were excluded. Note that drusen particles are early signs of aged macular degeneration, which can be treated in the same class as AMD.

### 4.2. Feature Weightings, Selection, and Classification

Principal Component Analysis (PCA) is one of the oldest and most widely used statistical techniques that reduces feature dimensionalities. PCA improves interpretability while preserving variability, which minimizes informational loss. This means finding new variables *F_i_* that are linear functions of those in the original features, while successively maximizing variance and maintaining uncorrelatability. This allows the representation of a class of features in the following form:$$\frac{\lambda_{1}}{\sum_{i=1}^{p}\lambda_{i}}F_{1},\;\frac{\lambda_{2}}{\sum_{i=1}^{p}\lambda_{i}}F_{2},\;\ldots \;\frac{\lambda_{p}}{\sum_{i=1}^{p}\lambda_{i}}F_{p}$$
where *λ_i_* eigenvalues are sorted according to the size as *λ_1_* ≥ *λ_2_* ≥ … ≥ *λ_p_* ≥ 0, *F_i_*, *i* = 1, 2, … *p*, features, *F*_1_ is the best feature and the 2nd, 3rd, …, *k*th (*k* ≥ 4) are other principal components.

The PCA used in this paper was based on Singular Value Decomposition (SVD), which was used to select our best features. *X* = *UΣV^T^* represents the SVD of an input matrix *X*, where *U* represents the reflection of *X*, the diagonal of Σ represents the variabilities, *σ*_1_, …, *σ_η_*, up to the *η*-th feature, *V^T^* is the ‘eigen vectors’ of *X*. *X* will be subtracted by its mean of each column of *X_mean_*, to obtain *X_m_* = *X* − *X_mean_*. This subtraction is commonly used for normalizing *PCA* data. We then calculated *X_PCA_*, which is defined by the following:
(5)$$X_{PCA}=V_{N\times N}{}^{T}X_{m,N\times M}{}^{T}$$
where *M* is selected based on the percentage of variability up to *η*, *η* ≥ *M*, *N* is the number of total observations in the dataset and the dimensions of *X_PCA_* is *N* rows by *M* columns.

Our feature selection also included feature weighting using neighborhood component analysis (NCA) Ref. [24] to refine our feature searching process. NCA is a feature weighting algorithm utilizing the nearest neighbor to maximize expected leave-one-out classification accuracy. Then NCA was optimized using stochastic gradient accent to learn a feature weighting vector. The output of the NCA is a vector, *ω*, of length of *η* representing the weight importance of each feature. Each element in *ω* is assigned to each column of matrix *X_PCA_*.

We then searched for sets of optimal features by reducing the number of columns of *X_PCA_* through iterations. The values of the weights in *ω* were used for removing features from *X_PCA_* at iteration. The weights can be written as the following:(6)$$\omega^{\left(l\right)}=\left(\omega_{1},\omega_{2},\ldots ,\omega_{M_{l}}\right),\eta \;\geq \;M_{l}$$
where *M_l_*, is the number of features remaining after some features have been removed at *l*th iteration. This idea was taken from Ref. [25] “Drop Out”, where a Neural Network with a large number of parameters tended to overfit a dataset, causing a decrease in performance. The function of dropout is to drop neuron units with low weights from the neural network during training after multiplying feature weights to feature vectors. Our algorithm utilized this idea to drop features with weights that were below a certain threshold. By dropping these unimportant features, we achieved faster training times and prevented overfitting. To multiply feature weights to feature vectors, ω can be rewritten in matrix form by replicating each element and putting them in column wise fashion. Ω^(*l*)^ is defined by the following:$$\Omega^{\left(l\right)}=\left[\begin{array}{ccc} \omega_{1} & \begin{array}{cc} 0 & \ldots \end{array} & 0\\ 0 & \begin{array}{cc} \omega_{2} & \ldots \end{array} & 0\\ \begin{array}{c} 0\\ 0 \end{array} & \begin{array}{cc} \begin{array}{c} 0\\ \ldots \end{array} & \begin{array}{c} \ldots \\ \ldots \end{array} \end{array} & \begin{array}{c} 0\\ \omega_{Ml} \end{array} \end{array}\right]$$
where Ω has *N* number of rows. We can rewrite our input classification matrix, *X_PCA_*^(*l*)^, in terms of Ω and at *l*th iteration using the following:(7)$$X_{PCA}{}^{\left(l\right)}=\left\{V_{N\times N}{}^{T}X_{m,N\times M_{l}}{}^{T}\Omega^{\left(l\right)},\tau_{\omega^{\left(l\right)}}>\omega^{\left(l\right)}\right\}$$
where $x_{i,PCA}{}^{\left(l\right)}$ is the feature vector in the *i*th row of $X_{PCA}{}^{\left(l\right)}$, “·” indicates point matrix multiplication operation, $\tau_{\omega^{\left(l\right)}}$ is the weight removal threshold at *l*th iteration, and $X_{m,N\times M_{l}}$ is the updated feature matrix with *M*_1_ number of features. The features that are dropped have feature weights below $\tau_{\omega^{\left(l\right)}}$ at the *l* iteration. *X_PCA_*^(***l***)^ is a new classification input matrix and is inputted to each of the classifiers. The classifiers used during each iteration are RUS-Boost Ref. [26], and Naïve Bayes Ref. [27] and support vector machine using Polynomial and Gaussian Ref. [28], Random Forest Ref. [29] and Adaboost Ref. [30]. $\tau_{\omega^{\left(l\right)}}$ values selected for this paper ranged from 0.1 to 1 with increments of 0.01. The classification accuracies, *a*(*l*,*k*), were recorded in each *l*th iteration at the *k*th classifier. *a*(*l*,*k*) was defined by the following:$$a\left(l,k\right)=C_{k}\left\{X_{PCA}{}^{\left(l\right)}\right\},\;k=1,\;2,\ldots ,\;6$$
where *C_k_* was one of the *k*th classifiers listed above. The maximum accuracy was found using the following:$$l_{max}=arg\;\underset{l,k}{\max}\left(a\left(l,k\right)\right)$$
(8)$$l_{max}=arg\;\underset{l,k}{\max}\left(C_{k}\left\{(V_{NxN}{}^{T}X_{m,N\times M_{l}}{}^{T}\Omega^{\left(l\right)},\tau_{\omega^{\left(l\right)}}>\omega^{\left(l\right)}\right\}\right)$$
where argmax(·) determines the maximum location or accuracy of *l*th at *k*th classifier. Dataset 1, 2, and 3 were trained in this fashion. The *η* was selected to be 100 for Dataset 1, and 70 for Dataset 2 and 3. Figure 4 shows the MSKξMP OCT Image recognition system from start to finish.

### 4.3. Performance Evaluation

To evaluate and determine the performance of the proposed feature extraction approach, the accuracy, sensitivity, specificity, precision, and F1-score were compared with the results of HOG Ref. [4], BACNN Ref. [7], Alpha Mean Trimmed Patterns Ref. [10], Fibonacci Patterns Ref. [19], Classical LBP Ref. [21], Vision Transformer Ref. [31], ResNet Ref. [32], VGG16 Ref. [33], and Inception V3 Ref. [34]. By definition, higher values on these indexes imply better quality measures of classification. Mathematical formulas of these measurements are given below:$$Accuracy=\frac{Tp+Tn}{Tp+Tn+Fp+Fn}$$
$$Sensitivity=Recall=\frac{Tp}{Tp+Fn}$$
$$Specificity=\frac{Tn}{Tn+Fp}$$
$$Precision=\frac{Tp}{Tp+Fp}$$
$$F1\;Score=\frac{2*precision*recall}{precision+recall}$$
where *Tp* and *Tn* are the true positive and true negative and *Fp* and *Fn* are the false positive and false negative, respectively.

## 5. Results and Discussions

For datasets 1 and 2, four different classification schemes were tested using MSKξMP, AMD vs. DME vs. Normal, AMD vs. DME, DME vs. Normal, Normal vs. AMD, and all classes. Since dataset 1 was slightly imbalanced, AMD (723 images), DME (1101), and has Normal (1407), using accuracy as the only measurement would not be enough to detail the efficacies of the MSKξMP. Using measurements such as recall, specificity, precision, and F1-score was more effective. For dataset 3, we also tested using four classification schemes: Normal vs. CNV, CNV vs. Drusen, Normal vs. Drusen, and all classes, Normal vs. CNV vs. Drusen. Our experiments included kernel sizes up to 3 × 9 and 5 × 9, however, these window sizes were too large, and important features detected may not be within the local areas of the pixel, *I_C_*. Therefore, kernel sizes up to *n* × 9 were omitted from the result section. As using more kernels with varying sizes would have diminishing returns and performance improvements would be minimal. Using six kernels is the right balance between computation times and performances of our MSKξMP.

### 5.1. Dataset 1, 2 and 3 Results

The highlights of our experiments on datasets 1 and 2 are shown in Table 2 and Figure 5, which show the performance measurements of our MSKξMP. The highest accuracy was achieved by SVM using polynomial kernel at 99.78% on dataset 1 and 96.59% on dataset 2. We achieved 100% on some of the binary classifications, Normal vs. AMD and Normal vs. DME, both using SVM with polynomials as well. Rus-Boost and Naïve Bayes seem to perform the worst, while Adaboost and Random Forest have similar performance to each other. This is because Adaboost and Random Forest both utilize an ensemble of weaker classifiers to create one stronger classifier: K-nearest-neighbor Classifiers to create Adaboost and decision trees for creating Random Forest. The highest accuracy was achieved by SMV polynomial. Regarding misclassifications, SVM polynomial kernel did not misclassify any normal images, only misclassified two DME class images, and using SVM Gaussian only two DME images are misclassified for dataset 1, see confusion matrixes Figure 5. Something to note on dataset 2’s results are their sensitivities (recall) and specificity, notice the specificity was 98.33%, which is higher than the accuracy and sensitivity; this suggests high confidence in their true negatives.

The main takeaway from the results in dataset 3 is that our MSKξMP was able to achieve 89.3% in accuracy using SVM polynomial kernel with specificity of 94.37%. This means high confidences in the true negative diagnoses. The other classifiers performed similarly to dataset 1 and dataset 2. Normal vs. CNV achieved the highest accuracy at 98.53% due to the significant differences between their visual features. It can be observed that MSKξMP was able to achieve 93.69% accuracy with the CNV vs. Drusen classification scheme. This is a good score because while CNV images contain drusen particles that may cause some confusion with AMD images, our feature extractor was able to differentiate between these two classes reliably.

Figure 6 shows ROC curves for our best results “SVM: Polynomial” in all three datasets. For each dataset, we plotted three ROC curves with one class being the positive class of the other two classes. For example, dataset 1 have three classes: Normal, DME, and AMD. One curve plots the True Positives vs. the False Positives using Normal as the positive class and DME and AMD as the negative classes. The area under the curves (AUC) are also computed for each respective curve. Notice the AUCs for dataset 1 were higher and the AUCs of dataset 3 were lower, which reflects their respective accuracies, sensitivities, and specificity.

### 5.2. Ablation Study

For our ablation study, we tested the feature extraction capabilities of each kernel size and weights (shown in Figure 3) on dataset 1. From Table 3, it can be observed that the accuracies measured were highest with the 5 × 7 and 7 × 7 kernels. For each kernel size, the accuracies were about 0.5–3% below our highest achieved accuracy of 99.78% (all kernels). Concatenating the feature vectors generated from each kernel size helped improve the performances of our MSKξMP and indicated that each sized kernel captured different textural information from the OCT image, both locally and regionally. Based on this ablation study, we decided to omit the 3 × 3 kernel and test combined features generated by 5 × 5, 7 × 7, 3 × 5, 5 × 7 and 3 × 7. We were able to achieve an accuracy of 99.78% by SVM polynomial kernel without using the 3 × 3 kernel.

### 5.3. Comparisons to Recent State of the Art

Comparing dataset 1 with recent state of the art shows our MSKξMP is able to achieve slightly higher accuracies, as shown in Table 4 and Table 5. Das Ref. [7]’s BACNN achieved an accuracy of 97.76% with a plus or minus of 2.24%, meaning BACNN was able to achieve 100% but with extremely high fluctuation. Even though our experiments were based on ten-fold validations, our fluctuation was only about 0.3–0.4%, which is more stable than BACNN. Compared to other state of the arts, our MSKξMP outperformed very deep CNNs such as Refs. [13,32,34] as well as the recent discovery of vision transformer algorithm Ref. [31]. Also, our MSKξMP was able to outperform recent state of the art texture descriptors such as Refs. [10,19], by 99.78% to 94.96% and 99.16% respectively, see Table 6. Table 7 shows datasets 2 and 3 comparisons between the recent state of the art and our MSKξMP. Our accuracies either outperformed or were comparable against deep learning techniques such as Ref. [35]. Note that Ref. [35] had high fluctuations across all performance measurements, ±0.8–3%, whereas our results with MSKξMP only fluctuated within 0.1–0.2%.

## 6. Conclusions and Future Work

This work presents a new image textural descriptor, MSKξMP, for differentiating between ocular diseases such as AMD, Drusen, DME, and CNV in OCT images. The presented method can be used to encode textural patterns at local and regional scales and to improve edges in various directions while avoiding speckle noises. The computer simulations show that our measurements outperformed (a) the current state-of-the-art deep learning techniques and vision transformer; and (b) the FPN-EfficientNetB0 in dataset 3 and is comparable to the FPN-ResNet50 network. Some of our binary classifications from dataset 1 also performed well, achieving perfect accuracy, or close to it. MSKξMP had high specificity, suggesting high confidence in their true negatives. Future work should be focused on constructing 3D retinal images, such as 3D OCT images, by extracting volume and depth information to determine the spread of the disease. The 3D structures should be used for multi-view classification.

## Figures and Tables

**Figure 1 diagnostics-13-00729-f001:**
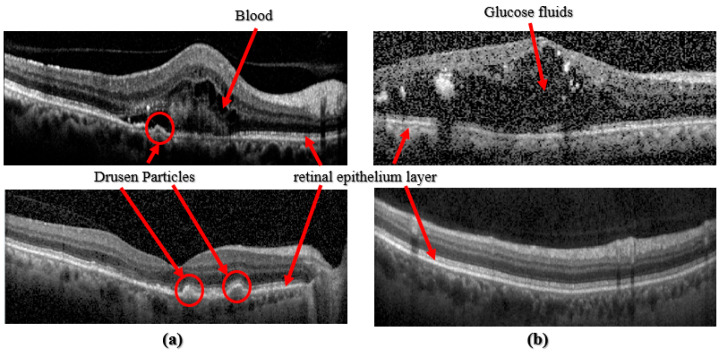
(**a**) Images taken from [8], top: CNV class, bottom: Drusen class; (**b**) Images taken from [9], top: DME class, bottom: Normal class.

**Figure 2 diagnostics-13-00729-f002:**
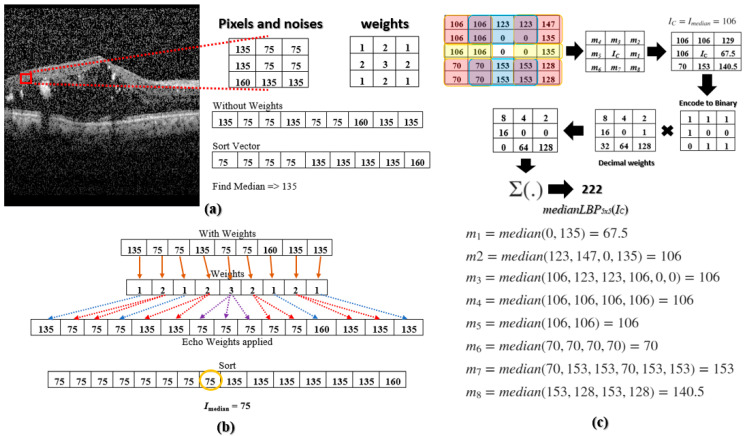
(**a**) A noisy DME OCT image taken from [9] with a 3 × 3 kernel of clean and noisy pixels; median is calculated; (**b**) Finding the median using echo-based weights; (**c**) A sample calculation of medianLBP5 × 5.

**Figure 3 diagnostics-13-00729-f003:**
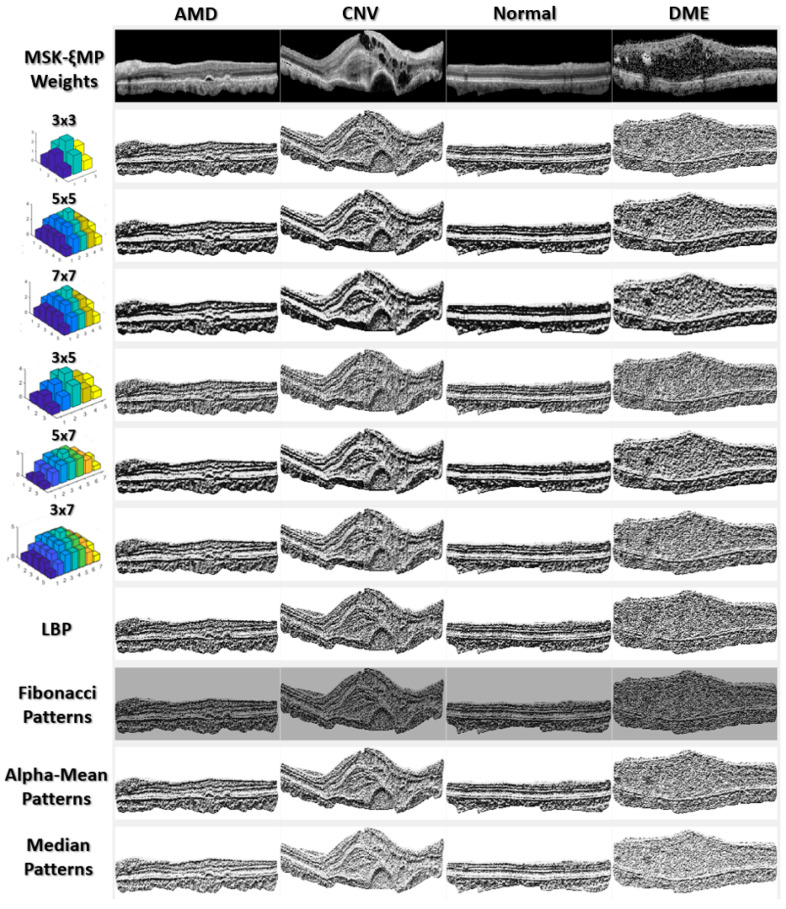
MSKξMP images associated with their weights, generated using Echo Formula.

**Figure 4 diagnostics-13-00729-f004:**
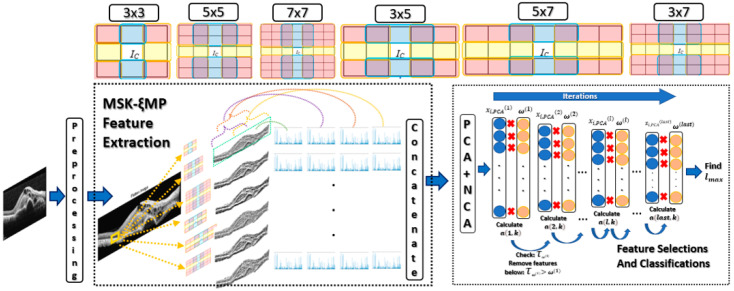
MSKξMP OCT Image recognition system.

**Figure 5 diagnostics-13-00729-f005:**
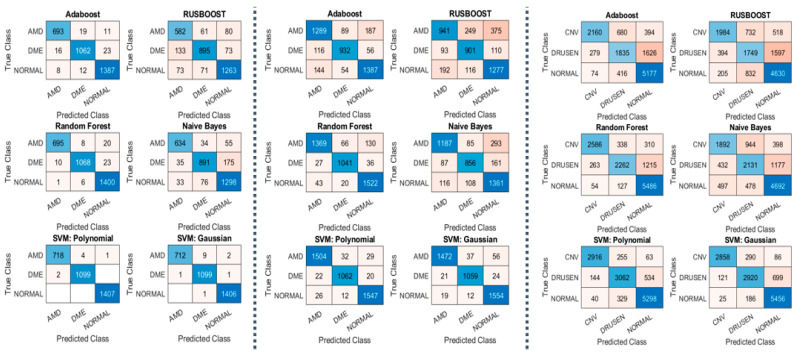
(**Left**) Confusion Matrixes from Dataset 1; (**Middle**) Confusion Matrixes from Dataset 2; (**Right**) Confusion Matrixes from Dataset 3.

**Figure 6 diagnostics-13-00729-f006:**
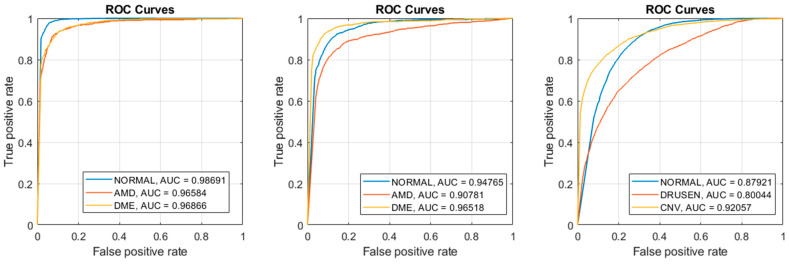
SVM: Poly produced the best results (**Left**) ROC curves of Dataset 1; (**Middle**) ROC curves of Dataset 2; (**Right**) ROC curves of Dataset 3.

**Table 1 diagnostics-13-00729-t001:** Shows prior works and their results.

Article	Dataset	Algorithm Used	Results
Srinivasan [4]	Duke Dataset: 15 normal subjects, 15 patients with Age-related macular degeneration (AMD) 15 patients with Diabetic Macular Edema (DME)	Preprocessing: Denoised, Flattened, and cropped image; Feature extraction: HOG descriptor using Image pyramid; Classifier: SVM	Normal 13/15 = 86.67% AMD 15/15 = 100% DME 15/15 = 100%
Sun [15]		Preprocessing: Retinal Alignment using polynomial regression fitting Feature Extraction: Dictionary Learning, Sparse Coding and Max Pooling Classifier: Linear SVM	Normal 14/15 = 93.33% AMD 15/15 = 100% DME 15/15 = 100%
Wang [16]		Feature Extraction: Spatial Pyramid Features, Multi-scale Spatial Pyramid Features; Classifier: SVM	AMD 0.978 ± 0.008, DME 0.940 ± 0.004 Normal 0.996 ± 0.001, Overall Acc 0.980 ± 0.001
Hussain [17]		Feature Extraction: 3D segmentation Canny Edge Detection, extracting curved boundaries using polynomials Classifier: Random Forest	2 Classes Normal vs. DME Sensitivity: 94.67%, Specificity: 100.00%, F1-score: 97.22, Accuracy: 97.33% 3 Classes Accuracy: 96.89%AUC: 0.99
Thomas [1]		Feature extraction: multi-Path CNN with residual connections Classifier: Sigmoid Fully Connected layers	Two Class AMD vs. Normal Accuracy: 96.67% AUC: 100%
Rong [14]		Preprocessing: Denoising and Flattening Feature Extraction: Surrogate-Assisted Retinal CNN	3-Classes—With Surrogates|With Raw Images AUC: 0.9856|0.9491 Acc: 0.9509|0.9205 Sen: 0.9639|0.9059 Spe: 0.9360|0.9371
Wang [18]		Preprocessing: Mean and Bilateral Filter and image cropping Feature Extraction and Classifier: CliqueNet	3-Classes Accuracy: 0.990 Precision AMD: 0.956|Recall AMD: 1 Precision DME: 1.00|Recall DME: 0.99 Precision Normal: 1.00|Recall Normal: 0.986
Das [7]		Feature Extraction: B-Scan Attentive CNN Classifier: Fully connected layer	3-Classes Acc: 97.12 ± 2.78, SP: 95.61 ± 4.35, Se: 97.76 ± 2.07, AUC: 0.97 ± 0.03
	Noor Eye Hospital dataset 148 volumes 48 AMD volumes (1565 total images) 50 DME volumes (1104 total images) 50 Normal Volumes (1585 total images)		AMD, Acc = 93.2 ± 2.7, SP = 95 ± 0.1, Se = 92.0 ± 4.4 DME: Acc = 99.3 ± 1.5, SP = 98.9 ± 2.4, Se = 100 Normal: Acc = 92.2 ± 2.3, SP = 93.2 ± 2.3, Se = 87.8±4.8
Wang [4]		Preprocessing: Flattening and Cropping Feature Extraction: ResNet50 with LSTM modules Classifier: Fully Connected layers	AMD 90.0 ± 7.1 94.6 ± 9.2 DME 94.0 ± 9.4 97.5 ± 5.6 Normal 98.0 ± 4.5 91.2 ± 5.9 Overall Accuracy: 94.0 ± 4.3 Overall Precision 94.4 ± 4.4 Overall Sensitivity 93.8 ± 4.8
Rasti [9]		Preprocessing: Retinal Flattening Feature Extraction: Multi-Scale Convolutional Neural Network Ensemble	3-Classes Highest Performances: l_3_ -- l_2_ -- l_1_ -- l_0_ Precision: 99.39 ± 1.21 Recall: 99.36 ± 1.33 F1: 99.34 ± 1.34, AUC: 0.998

**Table 2 diagnostics-13-00729-t002:** Results from datasets 1, 2 and 3.

Dataset 1						
3 Classes	AdaBoost	Naïve Bayes	SVM: Poly	R. Forest	SVM: RBF	Rus Boost
precision	0.9714	0.8809	**0.9976**	0.9806	0.9958	0.8340
sensitivity	0.9696	0.8696	**0.9971**	0.9754	0.9941	0.8385
specificity	0.9857	0.9317	**0.9989**	0.9885	0.9978	0.9240
accuracy	0.9725	0.8737	**0.9978**	0.9790	0.9957	0.8480
F-measure	0.9705	0.8740	**0.9974**	0.9779	0.9949	0.8354
**2 Classes**	**Normal vs. AMD**		**Normal vs. DME**		**DME vs. AMD**	
Classifier	SVM: Poly		SVM: Poly		SVM: Poly	
precision	0.9996		1.0000		0.9970	
sensitivity	0.9993		1.0000		0.9961	
specificity	0.9993		1.0000		0.9961	
accuracy	0.9995		1.0000		0.9967	
F-measure	0.9995		1.0000		0.9966	
**Dataset 2**						
**3 Classes**	AdaBoost	Naïve Bayes	SVM: Poly	R. Forest	SVM: RBF	Rus Boost
precision	0.8500	0.8066	**0.9662**	0.9256	0.9601	0.7347
sensitivity	0.8476	0.7975	**0.9663**	0.9260	0.9601	0.7410
specificity	0.9223	0.8977	**0.9833**	0.9615	0.9799	0.8655
accuracy	0.8481	0.8002	**0.9669**	0.9243	0.9603	0.7332
F-measure	0.8487	0.7997	**0.9663**	0.9249	0.9599	0.7326
**2 Classes**	**Normal vs. AMD**		**Normal vs. DME**		**DME vs. AMD**	
Classifier	SVM: Poly		SVM: Poly		SVM: Poly	
precision	0.9740		0.9877		0.9760	
sensitivity	0.9740		0.9870		0.9753	
specificity	0.9740		0.9870		0.9753	
accuracy	0.9740		0.9877		0.9764	
F-measure	0.9740		0.9873		0.9757	
**Dataset 3**						
**3 Classes**	AdaBoost	Naïve Bayes	SVM: Poly	R. Forest	SVM: RBF	Rus Boost
precision	0.7350	0.6730	**0.8931**	0.8343	0.8952	0.6608
sensitivity	0.6907	0.6609	**0.8851**	0.7908	0.8758	0.6327
specificity	0.8499	0.8385	**0.9431**	0.8985	0.9395	0.8191
accuracy	0.7256	0.6894	**0.8920**	0.8175	0.8887	0.6616
F-measure	0.7022	0.6652	**0.8888**	0.8026	0.8837	0.6414
**2 Classes**	**Normal vs. Drusen**		**Normal vs. CNV**		**Drusen vs. CNV**	
Classifier	SVM: Poly		SVM: Poly		SVM: Poly	
precision	0.9001		0.9817		0.9358	
sensitivity	0.8887		0.9782		0.9327	
specificity	0.8887		0.9782		0.9327	
accuracy	0.8993		0.9815		0.9345	
F-measure	0.8935		0.9799		0.9339	

**Table 3 diagnostics-13-00729-t003:** Results of ablation study.

Ablation Study						
Kernel Size	3 × 3	5 × 5	7 × 7	3 × 5	5 × 7	3 × 7
precision	0.9684	0.9832	0.9902	0.9772	0.9904	0.9746
sensitivity	0.9636	0.9808	0.9878	0.9743	0.9890	0.9734
specificity	0.9821	0.9920	0.9950	0.9872	0.9955	0.9868
accuracy	0.9666	0.9842	**0.9904**	0.9762	**0.9910**	0.9749
F-measure	0.9659	0.9820	0.9890	0.9757	0.9897	0.9739

**Table 4 diagnostics-13-00729-t004:** Dataset 1 comparisons to recent state of the art.

Dataset 1: Comparisons to State of the Art				
Author	Method	Sensitivity (%)	Specificity (%)	Accuracy (%)
DAS [7]	HOG+SVM [4]	88.47	96.48	90.83
	VGG16 [33]	94.80	91.15	93.73
	InceptionV3 [34]	91.46	98.22	93.46
	ResNet [32]	89.59	98.22	92.14
	BACNN	97.76	95.61	97.12
	**Our Work**	**99.71**	**99.89**	**99.78**

**Table 5 diagnostics-13-00729-t005:** Dataset 1 comparisons to recent state of the art.

**Author**	Thomas [3]	Srinivasan [4]	Hussain [17]	Wang [18]	Jiang [31]	Luo [36]	Mousavi [37]	Karri [38]	**Our Work**
**Accuracy**	96.66%	95.56%	96.89%	99.00%	99.69%	94.20%	98.38%	91.33%	**99.78%**

**Table 6 diagnostics-13-00729-t006:** Dataset 1 comparisons to recent texture descriptors.

Comparisons to Other Recent Texture Descriptors: Dataset 1				
Technique	AMT-LBP [10]	LBP [21]	Fibonacci [19]	MSKξMP
precision (%)	99.24	97.62	95.11	**99.76**
sensitivity (%)	98.95	97.01	94.60	**99.71**
specificity (%)	99.61	98.67	97.33	**99.89**
accuracy (%)	99.16	97.31	94.96	**99.78**
F-measure	99.09	97.30	94.84	**99.74**

**Table 7 diagnostics-13-00729-t007:** Dataset 2 and 3 comparisons to recent state of the art.

Dataset 2: Three Class Classification				
Author	Technique	Sensitive (%)	Precision (%)	Accuracy (%)
Wang [4]	OCANet-CK	88.50	89.40	88.60
	VOCT-RNN	94.00	94.40	93.80
**Our Work**	MSKξMP	96.63	96.61	96.68
**Dataset 3: Three Class Classification**				
**Author**	**Technique**	**Sensitive (%)**	**Specificity (%)**	**Accuracy (%)**
Sotoudeh-Paima [35]	FPN-EfficientNetB0	86.60	93.30	87.80
	FPN-ResNet50	89.80	94.80	90.10
**Our Work**	MSKξMP	88.51	94.31	89.20

## Data Availability

Duke Dataset is found here: https://people.duke.edu/~sf59/Srinivasan_BOE_2014_dataset.htm, NOOR datasets are found here: https://hrabbani.site123.me/available-datasets/dataset-for-oct-classification-50-normal-48-amd-50-dme and https://data.mendeley.com/datasets/8kt969dhx6/1 (accessed on 1 January 2020).

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
