# Peer review of "Distinctions between Choroidal Neovascularization and Age Macular Degeneration in Ocular Disease Predictions via Multi-Size Kernels ξcho-Weighted Median Patterns"

_diagnostics, 2023, doi:10.3390/diagnostics13040729_

Round 1
Reviewer 1 Report
See the attached report.

Reviewer 2 Report
Comments to the Author
In this paper, Distinctions between Choroidal Neovascularization and Age Macular Degeneration in Ocular Disease Predictions via Multi-Size Kernels ξcho-Weighted Median Patterns is presented. I have some major concerns about this paper:
1. Compared with end to end the deep convolutional neural Networks, the novelty of the proposed method is poor. Deep learning is not a black box.
2. The result analysis are insufficient. ROC should be presented in the experimental results
3. The citation level of references is in confusion.
4. The author's method still needs to select features manually. How can it be called automatic method?
5. There are many syntax errors.
6. Only some deep learning algorithms in 2014 and 2015 have been compared, which needs to be supplemented and compared with the latest deep learning algorithm.
Round 2
Reviewer 1 Report
I think that the authors have addressed all my previous remarks. The manuscript is suitable for publication in its present form.
Reviewer 2 Report
The author has basically revised it. I have no other suggestions.